# The Effect of Green Investment and Green Financing on Sustainable Business Performance of Foreign Chemical Industries Operating in Indonesia: The Mediating Role of Corporate Social Responsibility

**Jianmu Ye and Efifania Dela ***

School of Management, Wuhan University of Technology, Wuhan 430070, China; jianmuye@126.com
* Correspondence: delaefifania@gmail.com

**Abstract:** Emerging economies endeavor to achieve a green economy by realizing their potential for sustainable commercial success. Due to natural resource restrictions, businesses must concentrate on green investment, financing, and resources to promote sustainable company performance. To better understand how to implement corporate social responsibility (CSR) and sustainable company performance, this study looks at the effect of green financing and investment. The study used quantitative research techniques through primary and secondary data sources from Indonesia's 238 sampled international chemical companies. Additionally, a standardized questionnaire was employed in this study to gather data. The study used Smart-PLS and a structural equation model (SEM) to examine the data gathered and determine the relationship between green investment, green financing, CSR, and sustainable business performance. The study shows that green investments and financing significantly and favorably affect CSR and sustainable performance. Additionally, it was found that CSR significantly mediates green investment and green financing with sustainable business performance relationships. This work added to the body of literature and emphasized the significance of each construct. The study's conclusions also suggested that highly polluting chemical businesses should incorporate green financing, investment, and CSR to improve sustainable economic performance.

**Keywords:** green investment; green financing; CSR; sustainable business performance; stakeholder theory

## 1. Introduction

Highly polluting chemical industries in emerging economies face significant socioeconomic and environmental challenges. To address these challenges, companies can adopt green investment and green finance practices to promote environmentally sustainable practices. Additionally, companies can implement CSR initiatives to meet societal expectations and improve their reputation [1]. However, adopting green investment, green finance, CSR, and sustainable business performance practices has been given little attention in Indonesia's foreign industrial companies [2]. Foreign direct investment (FDI) by industrial companies has the potential to benefit the green economy by introducing new technologies, knowledge, and best practices that can improve the environmental performance of host country industries. FDI can also promote the growth of green industries and services, creating opportunities for sustainable economic growth and job creation. However, if not properly managed, FDI can also have negative environmental impacts, such as increased pollution and resource depletion, especially in countries with weak environmental regulations or enforcement. Host nations can create green investment frameworks and cooperate with multinational corporations to execute sustainable business practices that benefit businesses and the environment to promote sustainable development through foreign direct

investment (FDI) in industrial sectors. Implementing a green economy through green finance and investment is crucial for the economic progress of developing countries, and all stakeholders must recognize environmental challenges to promote sustainable growth [3]. Green finance offers a way to operate funds for the advantage of societal concerns, enabling various eco-friendly projects to achieve long-term sustainable performance [4]. This innovative financial instrument provides support for promoting the green transformation of the economy and plays a crucial role in promoting the harmonious development of the economy and environment. Investing in green and CSR initiatives can contribute significantly to sustainable financial performance [5].

Indriastuti and Chariri [6] investigating the effects of green and CSR investments on financial performance have yielded inconsistent results, leaving a gap in knowledge regarding how these investments can sustainably benefit a company's financial performance [6]. While some studies have shown that green investments can lead to financial gains, others have found no significant impact. Similarly, some studies have reported positive effects of CSR investments on stock value, financial performance, and other factors, while others have not [7]. Many organizations have recognized the potential advantages of incorporating CSR activities. Investing in CSR can enhance a company's reputation, profitability, and sustainable performance, improving its image overall [8]. However, some studies suggest that the high additional costs associated with social responsibility may not necessarily influence a company's financial or sustainable performance [9]. Despite the growing recognition of corporate social responsibility as an investment for growth and sustainable performance, previous studies have produced conflicting results regarding the relationships between green investment, green financing, corporate social responsibility, and sustainable business performance.

Furthermore, despite the increasing body of literature on the impact of green financing and corporate social responsibility practices on sustainable business performance, there is still a significant research gap that specifically investigates the large foreign industrial companies operating in Indonesia [10]. Previous studies have focused on developed and industrialized countries with higher pollution levels [11]. Chemical industries in developed nations produce various pollution that cause climate change, acid rain, and respiratory issues. Extraction, refining, and transportation of oil and gas release greenhouse gases, sulfur dioxide, nitrogen oxides, and particulate matter. Steel production and chemical manufacturing also cause pollution. Mining activities release heavy metals and particulate matter. Pharmaceutical manufacturing can harm aquatic life and cause antibiotic-resistant bacteria. Regulating and monitoring these industries is crucial to minimize their environmental impact on human health and the ecosystem.

Indonesia places great importance on the chemical sector as it is an industry that significantly contributes to the country's economy, providing job opportunities and generating revenue through exports. As per data from the Indonesian Ministry of Industry, the chemical industry accounted for approximately 4.8% of the country's GDP in 2019, employing over 2 million people. Additionally, due to its potential for high profits and growth, the chemical sector is often shown as an attractive industry for foreign investors, with foreign companies investing in Indonesia's chemical sector to take advantage of the country's abundant natural resources, strategic location, and low labor costs. Furthermore, Indonesia offers various investment incentives, such as tax holidays and import duty exemptions, making it an appealing destination for FDI. Given the chemical industry's substantial contribution to the economy and potential for further growth through foreign investment, it is likely that this sector is selected for research on FDI in Indonesia. Such research can provide valuable insights into the sector's competitiveness and attractiveness to foreign investors and help policymakers identify areas for improvement and formulate policies to encourage further investment.

However, the growing investment in foreign industries has grown in Indonesia, and the concern of green investment, green financing, and CSR practice have found a big gap. Therefore, more research is needed to fill this knowledge gap and provide a better under-

standing of the impact of green investment, financing, and CSR on sustainable business performance in Indonesia's highly polluted foreign industrial company's context. Therefore, based on the current debates on the above issues in green investment, green financing, CSR, and sustainable business relationships, the study emphasizes that the collaboration of policymakers and stakeholders is essential to promote sustainable development in Indonesia's foreign-polluting industries.

The selection of a particular area and subject of study is often influenced by various factors, such as the topic's relevance to the country, the availability of data, and the potential impact of the research results on the industry or society. For instance, our research on foreign chemical industries operating in Indonesia is likely due to the chemical industry's crucial role in the country's economy. This industry contributes significantly to Indonesia's GDP and provides employment opportunities for many Indonesians. In addition, the reason for selecting the chemical industry for a study on FDI in Indonesia may vary, but it is likely due to the sector's significant contribution to the country's economy, as well as its potential for further growth through foreign investment. Research on the chemical industry's FDI trends and patterns can provide insights into the sector's competitiveness and attractiveness for foreign investors, as well as help policymakers identify areas for improvement and formulate policies to encourage further investment.

Furthermore, the focus on green investment and financing in our study is likely due to the growing global concerns about environmental sustainability and the potential of businesses to promote sustainable practices. We aim to investigate the impact of such practices on the sustainable performance of foreign chemical industries operating in Indonesia and the mediating role of corporate social responsibility in this relationship. Our research aims to understand the potential benefits of green investment and financing for the chemical industry in Indonesia and how it can contribute to the country's sustainable development goals. Achieving sustainable development in such an industry requires stakeholders' active participation and collaboration to develop innovative solutions to balance economic, social, and environmental priorities. Hence, this research aimed to examine the effect of green investment and financing on sustainable business performance through the mediation role of CSR in Indonesia's foreign chemical industries. To achieve this aim, the researchers will focus on the following research questions:

1.  To what extent does green investment influence the sustainable business performance of foreign chemical industries in highly polluting industries of Indonesia?
2.  Does green financing contribute to the sustainable business performance of foreign chemical industry companies in Indonesia?
3.  Does CSR mediate the relationship between green investment, green financing, and sustainable business performance?
4.  Does CSR enhance the international chemical industries' sustainable business performance in Indonesia?

The study used a quantitative research methodology for data collecting and analysis. The study collected information from significant FDI industrial companies in Indonesia that produce large amounts of pollution. The study also collected and analyzed the data from a sample of FDI companies operating in this industry using a partial structural equation model through Smart-PLS, such as path analysis, to identify correlations between green investment, financing, CSR, and sustainable business performance of the foreign industrial companies in Indonesia.

The study has significant contributions to both academic and practical aspects. In the academic realm, the study's results can enhance the existing knowledge about the effectiveness of green investment and financing initiatives in FDI industries and shed light on the factors that may influence this relationship. The research can aid decision-makers and investors in adopting green and socially responsible investment strategies to increase profits while benefiting the environment. It can also assist investors in making investment decisions in sustainable companies. Moreover, the study can contribute to developing

effective CSR strategies for industries and inform policymakers about the most suitable strategies and regulations to promote sustainable financial activities in the region.

In addition, this research provides practical implications for companies operating in heavily polluting industries in Asia. Identifying the impact of CSR initiatives on business performance and the factors influencing this relationship can help companies develop more effective CSR strategies and identify the most appropriate CSR initiatives to implement in their specific context. It can aid companies in enhancing their reputation, brand value, and finances.

The article is set up as follows. Theoretical reviews and theories on the connection between green investment, green financing, CSR, and sustainable company performance are presented in Section 2, emphasizing research discussions. Section 3 provides a method description. The results of the study are shown in Section 4. The discussion and implications are presented in Section 5, and the final thoughts and suggestions for additional research are presented in Section 6.

## 2. Theoretical Background and Hypotheses Development

The stakeholder theory serves as the philosophical basis for this investigation. The stakeholder theory provides a theoretical framework for comprehending the connection between green investment, green financing, CSR, and sustainable company performance [12]. Businesses can achieve sustainable performance while adding value for society by considering the interests of all stakeholders, including the environment [13]. According to the stakeholder theory, businesses must consider all parties' interests, including their customers, suppliers, communities, and the environment [14]. This idea holds that businesses should consider their impact on the environment and greater society and optimize shareholder returns.

The industrial sector in Indonesia received foreign direct investment (FDI) in 2020, totaling USD 8.9 billion, despite a decrease of 6.9% compared to the previous year [15]. Major FDI projects from Japan, South Korea, Singapore, and China were noteworthy contributors to Indonesia's industrial sector FDI. However, foreign investors face obstacles such as a complex regulatory environment, a time-consuming and costly permit and license acquisition processes, underdeveloped infrastructure, concerns about labor quality and availability, employee-favoring labor laws, corruption, and growing economic nationalism [16]. The Indonesian government is implementing reforms to improve the business environment and attract more foreign investment [17]. Nevertheless, Indonesia faces challenges in green investment, financing, and corporate social responsibility, including a lack of knowledge and awareness among investors and the need for better regulatory frameworks and transparency in reporting. To overcome these challenges, stakeholders must collaborate toward a sustainable and responsible approach prioritizing transparency, accountability, and long-term value creation.

### 2.1. Green Investment

Green investment refers to investments that aim to mitigate the negative effects of human activities on the environment and promote the shift toward a sustainable, low-carbon, and resource-efficient economy [18] using a type of able energy, energy efficiency, and sustainable transportation. As climate change and sustainable development gain more attention, green investment is becoming increasingly important. Green investment is a crucial element in the pursuit of a sustainable and low-carbon economy, offering significant potential for environmental, social, and economic benefits. The investment in renewable power capacity surpassed that of fossil fuel-fired power capacity, highlighting the growing trend towards green investment and its potential to drive the transition towards a low-carbon economy [19]. Green investment not only has the potential to reduce greenhouse gas emissions and other environmental impacts but also to stimulate economic growth, create new jobs, and enhance human well-being [20].

"Green investment" refers to using public and private funds to purchase products and services that benefit the environment, such as preserving ecosystem diversity and repairing climate harm [21]. This investment serves society's three primary duties of preserving the environment, conserving resources, and upholding fairness and justice [22]. It is also morally right and consistent with an ecological civilization. The objective of a circular economy also incorporates green investment. It combines and coordinates the benefits of the economy, environment, and society to accomplish long-term social and economic development and create a harmonious society.

By minimizing their activities' negative environmental effects, such as energy use and carbon emissions, firms can win and keep the legitimacy and support of interested parties [23]. The annual report highlights the company's environmental concerns and reflects this strategy. In the end, society and its stakeholders decide the effects of green investment. Meanwhile, green investments contribute to a peaceful society that works toward the goal of the circular economy. By integrating and coordinating the positive impacts on the economy, environment, and society, the aim is to promote sustained social and economic development. Green investments can also help businesses gain and maintain the confidence and support of stakeholders. This is achieved by reducing energy consumption and greenhouse gas emissions, which minimizes the negative impact of the company's operations on the environment. The company's annual report shows its commitment to environmental sustainability. Furthermore, society and its members influence the outcomes.

### 2.2. Green Financing

Green financing encompasses financial tools and methods that are intended to facilitate environmental sustainability and facilitate the shift toward a low-carbon economy [24]. It involves the utilization of financial resources to support undertakings that have a positive impact on the environment, such as sustainable transportation, green buildings, renewable energy, and energy efficiency [25]. Green bonds are one of the most prevalent forms of green financing, which are debt securities offered by governments, financial institutions, and companies to finance projects that support the environment. The Climate Bonds Initiative defines green bonds as "debt instruments that raise funds for projects with specific environmental [26] benefits". Green loans are another type of green financing that is specifically created to fund green projects and activities. These loans can be utilized for a broad range of activities, from renewable energy projects to energy-efficient buildings. According to the Loan Market Association, a green loan is "any loan instrument made available solely to finance or refinance, in whole or in part, eligible Green Projects, whether new or existing". Other forms of green financing include green equity, green grants, and green insurance, which aim to promote environmental sustainability and support the transition towards a low-carbon economy. As the urgency to address climate change and environmental degradation intensifies, green financing is gaining traction among investors and financial institutions worldwide. Sustainable and responsible investments' total assets under management are on the rise, indicating the growing popularity of green financing. In summary, green financing pertains to financial instruments and methods that are designed to promote environmental sustainability and facilitate the shift towards a low-carbon economy. While green bonds and green loans are two popular forms of green financing, other types of green financing exist.

The green finance theory aims to connect the financial industry with the environmental protection industry to promote economic expansion while preserving the environment's natural resources. A comprehensive framework promotes sustainable development and directs investment in the environmental sector. By investing financial market funds in the environmental protection sector, green finance strives to connect the green economy with traditional finance [27]. Green finance supports investment and funding activities for sustainable human development, lowering environmental hazards, and promoting improving environmental quality. To achieve coordinated development among resources,

the environment, the economy, and society, the financial sector must act following environmental protection and sustainable development principles [28]. Green finance has the potential to accelerate the transition to a low-carbon economy and mitigate the impacts of climate change by providing financial incentives for companies to adopt sustainable practices. According to a study by the Global Commission on the Economy and Climate, green finance can "mobilize private capital at scale towards climate-friendly investments and activities" [29]. Furthermore, integrating environmental considerations into their business strategies can help companies reduce their exposure to regulatory and reputational risks and improve their operational efficiency and competitiveness in the marketplace [30]. This can lead to improved long-term financial performance for companies, as evidenced by a study by Eccles and Serafeim [31], which found that companies that prioritize sustainability outperform their peers financially. Therefore, green finance can play a critical role in driving the transition to a more sustainable economy [32], and it is not limited to supporting companies with low energy use only. The government supports the economy's environmentally and economically sustainable development by creating a system to encourage the development of green finance [33].

Green financing benefits long-term economic and social progress, particularly as China shifts to a low-energy and low-pollution mode of production [3]. Green finance promotes establishing an efficient industrial model and a sustainable energy structure, broadens the scope of business financing options, and reduces operational risks. By promoting the development of energy-saving and environmental protection technologies, the transformation of energy structures, and the implementation of energy-saving and emissions-reduction measures by businesses, green finance contributes to the restructuring of China's economic system [34]. Financial products such as green bonds, village funds, equities, and banking loans are included in green finance. It generates innovative financial solutions, offers better loan terms for sustainability initiatives, and increases markets by distributing knowledge about the advantages of sustainable projects [35]. Through market-oriented financial instruments such as green credit, green finance is essential for fostering the economic transformation toward more environmentally friendly and green economic growth [36]. The level of development in green finance is evaluated by considering the use of green financial instruments, the progress made in environmental improvement, and the emphasis placed on such development by local governments [37].

### 2.3. Sustainable Business Performance

A holistic perspective and a sustainability-focused strategy have been introduced to company development topics due to the growing concern for the quality of life of future generations [38]. A single or consolidated perspective would produce unreliable results for evaluating an organizational performance, which should be multidimensional [39]. According to academic studies by Fernando and Jabbour [40], three primary dimensions should exemplify a holistic picture of sustainable company performance: economic, social, and environmental value [41].

Sustainable business practices have become a crucial area of interest for current and future stakeholders due to their ability to ensure the long-term viability and well-being of the business and its associated economic, social, and environmental systems [42]. Despite this, many companies face the challenge of shifting their focus from traditional financial performance objectives to a more strategic approach that includes social and environmental sustainability [43]. Three essential performance criteria must be considered if business sustainability is to be achieved. The ability of the company to meet its present and future obligations is reflected in the first factor, financial sustainability. The second is social sustainability, which entails providing for people's needs and preserving positive social ties over time [44]. The third is environmental sustainability, which is concerned with preserving and regenerating the biosphere for future generations.

## 2.4. Corporate Social Responsibility (CSR)

A crucial component of business operations is CSR, which involves a corporation making voluntarily positive environmental contributions through investments in the market, the environment, and society [45]. The business is accountable for its decisions that impact consumer behavior, society, and the environment. CSR is pursuing financial success while upholding moral principles, society, and the environment. Corporate business goals to improve society's environment have benefited from CSR over time [46]. Although there are several methods of CSR, some businesses feel that their main duty is to increase shareholder value. Businesses should still be concerned with societal well-being despite creating riches for shareholders. Enterprises and society are interconnected stakeholders; whereas society depends on enterprises for services, infrastructure investment, and economic growth, businesses depend on society for resources and lower demand [47]. As a result, connections exist between the environment, society, and economy.

Businesses have recently switched their focus away from a fragmented strategy and toward a complete approach in the era of sustainable development. CSR has been crucial in emerging markets such as Indonesia, where social and economic issues are pervasive, and business ethics are lax. Incorporating socio-environmental issues into business operations is made possible by CSR, which also sets the standard for enterprises to prioritize safeguarding their local communities' natural, social, and economic surroundings [48]. This denotes that a corporation has strategically raised its major investment in CSR, allowing it to be integrated with macro marketing initiatives. Micromarketing's main objectives are to increase economic value and satisfy societal needs. Companies can combine social initiatives, environmental stewardship, and shareholder value development by integrating green marketing strategies into macro marketing efforts, bridging the gap between two different CSR viewpoints. As companies strive to enhance the environment, there has been an increase in customer loyalty, improved green brand positioning, and profitability, owing to the growing demand for eco-friendly products.

The study discusses how socially responsible businesses address societal needs, enhance their reputation, and ensure lasting product demand. CSR is a term that refers to a company's actions that impact stakeholders, the environment, the economy, and society. Using CSR as a bridge, the study focuses on the effects of green investment and green financing on the sustainable business performance of foreign chemical industries in Indonesia. The study seeks to understand how CSR functions as a mediating factor between green investments and green financing and these companies' sustainable business performance. The article underlines how many businesses have adopted CSR as one of their core values because it substantially affects people, the environment, and profit.

## 2.5. Rights and Obligations of the Manager in Companies

Managers are defined as individuals who are responsible for making decisions and managing resources in companies. They are key stakeholders in implementing green investment and financing strategies and integrating CSR into business operations, as they are responsible for allocating resources and making strategic decisions that influence the company's sustainability and performance. According to Robbins and Coulter [49], managers have the right to make decisions about the company's operations, access information about its financial performance, and receive compensation for their work. They also have the obligation to act in the best interests of the company and its stakeholders, comply with relevant laws and regulations, and act with due care and diligence. In the current study of foreign chemical industries operating in Indonesia, managers have additional rights and obligations related to implementing green investment and financing strategies, promoting corporate social responsibility, and ensuring sustainable business performance [50]. These include the right to allocate resources to green investments and financing, promote corporate social responsibility, and make decisions that prioritize long-term sustainability. Managers also have the obligation to comply with environmental regulations and stan-

dards, engage with stakeholders and respond to their concerns, and monitor and report on the company's sustainability performance [51].

The number of executives in a foreign chemical company can influence the level of decision-making authority that a manager holds. A larger number of executives may limit the manager's autonomy, while a smaller number may require the manager to take on more responsibilities. The qualifications and commitment of the manager are also important factors that can influence their rights and obligations in the company. Managers must have relevant education, experience, and skills, as well as a commitment to the company's goals and values, to avoid legal and financial liabilities and effectively fulfill their roles and responsibilities. It is crucial for managers to understand these factors to succeed in their roles. According to Funta [52], the size of a company determines the number of executives required, with larger companies needing more executives to ensure flexibility during normal operations. However, having a larger number of executives also increases the risk of unfavorable contracts being signed. Gregusova et al. [53] partially disagree with this view, stating that a company with only one manager can also make such contracts, but having fewer managers can be easier to control for partners. The rights and obligations of managers in foreign chemical companies are influenced by several factors, including the articles of association, the number of executives, and the qualifications and commitment of the manager. It is essential for managers to have a clear understanding of these factors to fulfill their roles and responsibilities effectively and avoid any legal or financial liabilities.

### 2.6. Theoretical Relationships of Green Investment, Green Financing, CSR, and Sustainable Business Performance

### 2.6.1. Green Investment and Sustainable Business Performance

Several theoretical frameworks explain the link between green investments and long-term success. The stakeholder approach contends that businesses must consider the interests of all parties, including the environment when making decisions [54]. According to the resource-based perspective, businesses can gain an edge over rivals by building special resources and competencies, such as environmental management systems, that benefit all stakeholders. Finally, the institutional theory suggests that companies may adopt green practices to conform to societal norms and expectations [55]. Empirical studies have investigated the correlation between green investment and sustainable performance in various contexts, including industries, countries, and periods. Hence, these studies have used different measures of green investment, such as investment in renewable energy, carbon emissions reduction, and environmental management systems [56].

Many research studies have explored the relationship between green investment and sustainable business performance. However, the majority of these studies have primarily concentrated on emerging and developed countries. Despite the fact that many multinational corporations have invested in various parts of the world during the recent era of industrialization, research in this area has been neglected [57]. The findings of the studies demonstrate a positive correlation between green investment and sustainable performance, suggesting that organizations that adopt green technologies and practices are more likely to attain sustainable performance outcomes. Additionally, various studies have reported a positive link between green investment and sustainable performance. A study by Eccles and Serafeim [58] revealed that companies that disclose their environmental performance achieve superior financial performance compared to those that do not. Another study found that sustainable tourism practices could improve environmental and economic performance [59]. Nevertheless, some studies have produced mixed or negative findings. The study conducted by Kassinis and Vafeas [60] reported that investing in environmental management systems did not significantly influence financial performance. Similarly, another study found that investing in renewable energy had a negative effect on financial performance in the short term but produced positive results in the long term [61].

The study also finds that this relationship is mediated by environmental management practices, suggesting that companies that adopt strong environmental management

practices are more likely to achieve sustainable performance outcomes due to green investment. The study identifies specific green investment strategies associated with sustainable performance outcomes, including eco-friendly product development, energy efficiency improvements, and waste reduction initiatives [62]. These findings suggest that companies prioritizing these green investment strategies are more likely to achieve sustainable performance outcomes. Thus, it provides evidence that green investment can positively influence sustainable performance outcomes, particularly when combined with strong environmental management practices [63].

**H1.** *Green investments and sustainable business performance are positively correlated.*

2.6.2. Green Investment and CSR Relationships

As companies increasingly recognize the importance of sustainability and environmental responsibility, green investment and CSR concepts have become closely related [6]. Empirical studies have shown that companies with higher levels of CSR engagement are more likely to invest in renewable energy projects and receive green loans designed to finance environmentally friendly projects. Conceptual research has proposed frameworks and models to understand the linkages between CSR and green investment, identifying factors such as institutional support for sustainability, availability of green investment opportunities, and financial resources [64]. These studies suggest that companies that engage in sustainable practices and CSR activities are more likely to have access to green investment opportunities and may benefit from improved reputation, reduced risk, enhanced stakeholder relations, and increased access to capital. Overall, the research indicates a positive relationship between green investment and CSR.

**H2.** *Green investment has a positive significant effect on CSR practice adoption.*

2.6.3. Green Financing and Sustainable Business Performance

Walley and Whitehead's neoclassical argument from 1994 claims that environmental legislation increases corporate costs [65]. According to their argument, reducing pollution would increase production costs and reduce marginal net profit if marginal costs continue to rise. In other words, green firms benefit from green financing as it positively impacts their performance [65,66]. According to Porter, effective environmental legislation should encourage businesses to innovate more because doing so will increase productivity, reducing the cost of protecting the environment and boosting businesses' profitability. Scholars are beginning to question the neoclassical theory. The stakeholder theory and natural resource-based view are the two theories that best describe the situation. The natural resource-based approach maintains that businesses can employ organizational resources to address environmental issues by integrating environmental elements into the RBV theory's research framework. It gives businesses a competitive edge while also enhancing financial performance.

The stakeholder theory states that businesses can adapt to changing external environments by meeting stakeholder needs and increasing organizational effectiveness. This improves corporate reputation and fosters the development of long-term relationships with suppliers and clients, which boosts financial performance [67]. In other words, the performance of green enterprises is positively impacted by green finance. Numerous researchers confirmed the results. Deng and Lu [68] stated that green finance practices could enhance the environmental performance of food companies. The environment and economy are greatly impacted by green financing, according to a study by Zhou et al. [28]. In light of the data above, this study suggested the following idea:

**H3.** *Green finance has a significant positive impact on sustainable business performance.*

2.6.4. Green Financing and CSR

Green financing and CSR are two linked ideas that have attracted much attention lately. CSR is the term used to describe a company's obligation to society and the environment and

its legal and financial responsibilities. Green funding describes the financial goods and services created especially to promote environmentally friendly and sustainable projects [69]. The relationship between CSR and green financing is that both aim to promote sustainable development [70]. Companies that adopt CSR policies are more likely to be interested in financing green projects, as it aligns with their values and can boost their reputation as socially responsible organizations [71]. Likewise, green funding can encourage companies to invest in environmentally friendly infrastructure and technologies.

Gangi et al. [72] revealed that green financing could positively impact a company's environmental performance and CSR practices. Another study by Wang et al. [73] examined the relationship between CSR and green bonds, a fixed-income security specifically issued to fund environmentally friendly projects. The study found that companies with better CSR performance are more likely to issue green bonds to enhance a company's CSR reputation. The relationship between CSR and green financing is mutually beneficial, as they both promote sustainable development and can enhance a company's reputation as a socially responsible organization. Green financing can encourage businesses to embrace sustainable practices and invest in environmentally friendly technologies and infrastructure. Companies that follow CSR policies are more likely to be interested in financing green initiatives.

**H4.** *Green financing positively impacts the adoption of CSR practices.*

2.6.5. Corporate Social Responsibility and Sustainable Business Performance

A study by Mani et al. [74] has established a positive correlation between CSR and sustainable company performance. The findings indicate that organizations that embrace CSR principles are more likely to achieve successful outcomes in sustainability. The study further suggests that the association between CSR and sustainable business performance is more robust in developed nations, implying that CSR practices promote sustainability in countries with well-established institutional support for CSR. Moreover, empirical studies have confirmed the correlation between CSR and sustainable performance. CSR and financial performance were found to have a marginally positive association, according to research done [75].

In contrast, a study by Kolk and Pinkse [76] found that companies with higher levels of CSR engagement were more likely to develop innovative products and services that addressed environmental and social issues, leading to improved financial performance. The study also identifies CSR practices associated with sustainable performance outcomes, including environmental management, employee relations, and community involvement [77]. These findings suggest that companies prioritizing these CSR practices are more likely to achieve sustainable performance outcomes.

**H5.** *CSR significantly influences sustainable business performance.*

*2.7. Mediating Role of CSR*

2.7.1. Green Investment, CSR, and Sustainable Business Performance

Through CSR and green investment programs, which improve financial performance and sustainability and aid in avoiding legitimacy gaps and social and environmental conflicts [6], businesses can demonstrate their accountability to stakeholders. Activities related to environmental accounting can improve a business's standing and credibility, which has a lasting positive effect. Compared to businesses with poor environmental performance, those with strong environmental activity typically have a broader shareholder base, more risk-sharing options, and better share values [78]. Companies utilize green investment to support stakeholders and to develop, and retain credibility. According to studies, green investments are positively correlated with a company's financial and sustainable performance [79].

However, CSR investments depend on intangible assets such as innovation, human capital, reputation, and culture [67]. It entails CSR environmental preservation efforts

that have received government, community, and shareholder approval. CSR investments strive to create positive relationships between the business and the environment by addressing the organization's economic, social, and environmental consequences. According to studies, CSR investments can boost a company's performance and reputation, luring stakeholders and encouraging them to become business owners by purchasing shares [77]. It is possible to view a favorable link between CSR investment and financial performance as a way to boost financial gains. Financial performance is eventually improved by effective environmental management. Implementing green investments and CSR investments can help businesses protect the environment, advance social welfare, and grow the economy, according to a sustainability report with green investment standards [80]. This is done by enhancing governance and stakeholder relations, enhancing reputation, and fostering trust. A positive stakeholder connection can improve financial performance, productivity, sales, and profitability.

The support of the company's stakeholders is essential, and the level of financial performance influences future investment choices. As part of a corporate strategy, managers must decide whether to invest in green and CSR efforts because sustainable performance is balanced based on people-planet-profit [81]. The easing of economic and social aims toward higher social and financial performance is achieved by integrating such activities into business operations [82]. A company's financial performance influences sustainable performance, and integrating green investments and CSR investments can boost a company's financial performance, improving sustainable performance [83].

**H6** : *CSR mediates significant on the relationship between green investment and sustainable performance*

2.7.2. Green Financing, CSR, and Sustainable Business Performance

Green financing gives money to projects that benefit the environment, such as resource efficiency and sustainable financial practices. CSR refers to a company's voluntary endeavors to tackle social and environmental problems that go beyond its legal requirements [84]. Sustainable performance refers to a company's ability to maintain long-term profitability while minimizing negative environmental and social impacts. Therefore, the study revealed that green financing and sustainable performance are closely related. Companies that obtain green funding typically perform better in terms of the environment because they can fund initiatives that lessen their environmental effect and increase resource efficiency [85]. Similarly, companies that actively engage in corporate social responsibility activities tend to demonstrate improved environmental and social performance, as they can establish trust with their stakeholders and enhance their reputation [86].

The study by Gilchrist et al. [87] has also demonstrated a positive correlation between corporate social responsibility and green financing. Companies with a robust CSR reputation are more likely to attract green financing, as investors are more interested in supporting companies committed to sustainable practices. This is because green financing is often seen as a way to promote sustainable development and address environmental and social issues. One argument is that CSR practices can mediate between green financing, investment, and sustainable business performance [88]. This means businesses can enhance their access to green financing and investment by engaging in CSR practices, leading to improved sustainable business performance. CSR practices can positively impact a company's financial performance, which can be further augmented by leveraging green financing and investment opportunities. Companies prioritizing CSR initiatives are more likely to achieve long-term financial success [89]. Using green financing and investment can further augment this success. Another argument is that green financing and investment can promote the adoption of sustainable business practices, enhancing a company's CSR profile. This idea is supported by research showing that companies that engage in sustainable business practices are more likely to be viewed positively by stakeholders, including investors and customers [90]. CSR practices can enhance access to green financing and investment.

The argument is that engaging in CSR practices can improve businesses' access to green financing and investment, leading to better sustainable business performance. Investors are increasingly interested in funding socially responsible and sustainable projects, highlighting the growing importance of sustainability in investment decisions. Research has shown that CSR can mediate between green financing, investment, and sustainable business performance [89]. Studies have also found a positive link between CSR practices and green loans, with stronger relationships for companies with higher environmental performance. While there is a need for further research and analysis, evidence suggests that CSR practices can enhance the effect of green investment and financing on sustainable business performance [91].

**H7.** *CSR significantly influences green financing and sustainable performance relationships.*

The study also suggested a research model that employs CSR as a mediator variable to investigate the impacts of green financing and investment made by foreign industrial corporations on the performance of sustainable businesses. Green investment and financing are included in the model as independent factors, whereas sustainable business performance and CSR are included as mediator and dependent variables, respectively. All study variables are represented in the conceptual model in Figure 1.

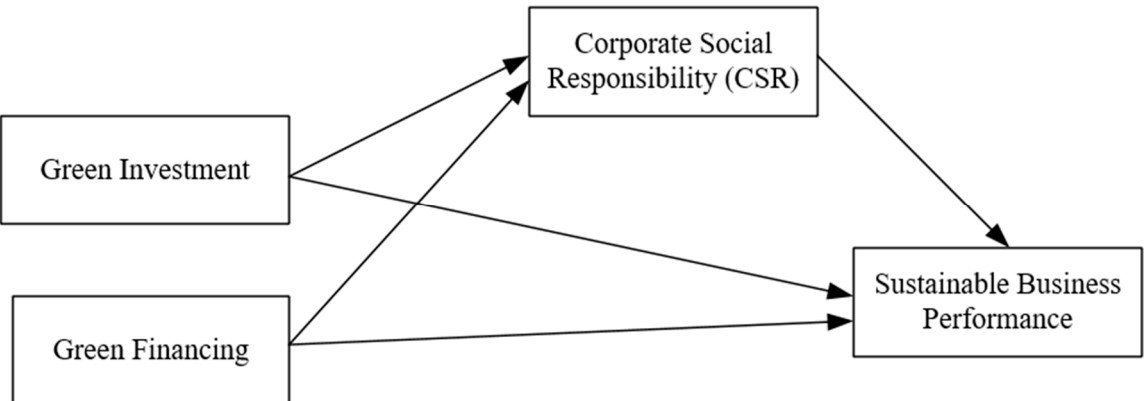

**Figure 1.** Conceptual Framework model.

## 3. Methodology

### 3.1. Design of the Study

The study aims to investigate the relationship between green investment and green financing on the sustainable business performance of foreign chemical industries operating in Indonesia. Additionally, the study aims to examine the mediating role of corporate social responsibility in this relationship. The study likely used a questionnaire to collect data from foreign chemical industries operating in Indonesia. The questionnaire included items measuring green investment, green financing, corporate social responsibility, and sustainable business performance. The collected data have been analyzed using Smart PLS statistical methods to examine the relationships between these variables and to test the mediating role of corporate social responsibility of foreign chemical industrial companies in Indonesia using a deductive reasoning technique and quantitative research methods [92]. The independent factors in our analysis were green financing and investment. The model also addressed long-term business performance, with corporate social responsibility as a study mediator variable. The study was conducted from August 2022 to January 2023. The authors of the study administered questionnaires to the participants. The questionnaires included measures of green investment, green financing, CSR, and sustainable business performance. The data collected in the study is not currently publicly available. The study was not commissioned by any external organization. Researchers (Students) at Wuhan University of Technology conducted it. Data from both primary and secondary sources were used in our study. The original data was collected from general managers

and marketing directors of industrial companies, whereas company sustainability reports, annual government reports, and policy documents were collected as secondary data. The study included 20 standardized questionnaires formatted in a Likert scale fashion, with a score of 1 representing the strongest disagreement and 5 representing the strongest agreement.

### 3.2. Sample and Techniques

The study used purposive sampling to select a sample of 400 businesses, of which 250 industrial companies were specifically chosen based on their current CSR activities, experience, and accessibility to data [93]. Consequently, out of the 400 foreign industries in Indonesia, the 250 industrial companies were specifically picked based on their current operational activities on CSR, experience level, and accessibility to the necessary data. Then, specific respondents were selected via a critical sample approach since management teams believed to be more knowledgeable about the company's financial situation than regular personnel [94]. Before the data collection, all respondents were made aware of the purpose of the study. A self-administrated questionnaire was given to each company (general manager, vice manager, finance manager, and regular to collect the necessary data). Due to missing information from the 238 responses delivered to respondents, 12 surveys were disqualified. Finally, additional analysis was conducted using these surveys. More than 10% of the population must be accurately sampled to be representative [95]. As a result, 60% of the industrial companies in the study region are included, as opposed to the recommended 10%.

### 3.3. Data Analysis

The study used Smart-PLS to analyze the relationships between the constructs. PLS-SEM is a second-generation regression model with two unique parts [96]. A measuring model, which made up the first part, was utilized to assess the dependability and accuracy of numerous indicators. The study used the structural model in the second analysis stage to test the hypothesis and identify the structural relationships among the variables [97]. The study's mediation analysis also employed PLS-SEM to determine the direct and indirect effects [98].

### 3.4. Measurements

The study employed various techniques to measure the variables of interest. A Likert 5-point scale, ranging from 1 (indicating strong opposition) to 5 (indicating strong agreement), was utilized to create the questionnaire. The study encompassed four different variables in the conceptual frameworks, consisting of a total of 20 measuring items, to comprehensively capture all independent, dependent, and mediator variables.

#### 3.4.1. Dependent Variable

In our study, "sustainable business performance" was defined as the degree to which a business operates in an environmentally and socially responsible manner while maintaining financial performance. To measure this construct, we used a composite index that included several indicators, such as energy efficiency, waste reduction, employee satisfaction, customer satisfaction, and financial performance. These indicators were selected based on their relevance to sustainable business practices and were weighted based on their importance as determined by a panel of experts in the field. To measure, we used a combination of self-reported data from the participating businesses and publicly. The dependent variable in this study was sustainable business performance. Results that had already been independently confirmed [40] were adjusted to create new measurement criteria for this variable. These metrics include sales volume, market share, profitability, and customer happiness. Appendix A (Table A1) contains the surveys with contextual information.

### 3.4.2. Independent Variable

The researchers employed green finance and investment as independent variables that were quantified using green investment and financing indicators created [99,100]. The measurement criteria, which conceptualize green investment and financing, include the degree to which CSR practices are applied to sustainable business performance, the amount of green funding allocated to sustainable performance, the availability of suitable corporate structures to manage all corporate operations, and the level of environmental sustainability in participating community development initiatives.

### 3.4.3. Mediator Variable Measurement

This study used CSR as a mediator variable to meet its goals. To what extent CSR mediates the links between green investments, green financing, and sustainable company performance was determined using the observed variables. The mediating variable items were used to examine how CSR activities impact green investments, finances, and sustainable business performance in chemical industries.

## 4. Results and Analysis

### 4.1. Demographic Data

According to the demographic information of the 238 sample respondents in the current study, 72% of respondents were male and 28% were female, respectively. According to statistical evidence, female engagement is little. This could result from companies being hesitant to offer female employees more credit. Most respondents' age ranges were within a range for the age group. It suggests that most respondents were responsible adults who could understand the researcher's questions and give proper answers. Regarding the respondents' educational backgrounds, the study's findings indicate that 31.2% had a bachelor's degree, 52.9% had a master's degree, and 15.9% had a doctorate. Table 1 suggests that most respondents could comprehend and contribute helpful information for the study.

**Table 1.** Demographic Analysis.

| Demographic Variables | | Frequency | % |
|---|---|---|---|
| Gender | Male | 172 | 72 |
| | Female | 66 | 28 |
| Age | 20–29 | 16 | 6.6 |
| | 30–39 | 58 | 24.4 |
| | 40–49 | 118 | 49 |
| | ≥50 | 48 | 20 |
| Education | Doctorate | 38 | 15.9 |
| | Masters | 126 | 52.9 |
| | Bachelor | 74 | 31.2 |
| Experiences (Year) | 5–10 | 37 | 15.6 |
| | 11–15 | 77 | 32.2 |
| | 16–20 | 104 | 43.8 |
| | >20 | 20 | 8.4 |
| Managerial Level | General Manager | 53 | 22.3 |
| | Financial Manager | 59 | 24.8 |
| | Regulators | 73 | 30.6 |
| | Corporate ambassadors | 53 | 22.3 |

Source: Survey data (2023).

### 4.2. Measurement Model Analysis

A crucial element of the model used to investigate the connection between latent variables and their measures is the measurement model [101]. Its goal is to evaluate the

constructs' reliability and validity. The measurement model in PLS-SEM is the starting point for assessing the precision and consistency of the indicators.

### 4.2.1. Reliability and Validity Test

This study evaluated the validity and reliability of each relevant questionnaire item using PLS-SEM statistical methods. The measurement model was assessed using factor loadings, Cronbach's alpha, composite reliability, convergent validity, and discriminant validity [102]. Composite reliability and Cronbach's alpha were used to evaluate the scale items' internal consistency [103]. While Cronbach's alpha and composite reliability should normally be higher than 0.70, AVE and factor loadings should also be greater than 0.50 [104]. The Cronbach's alpha and CR values above 0.70, as shown in Table 2 and Figure 2, demonstrate the reliability and robustness of the sample data used in this investigation. Significant AVE and factor loading values above 0.50 were also noted, pointing to the items' high correlations with genuine convergent validity.

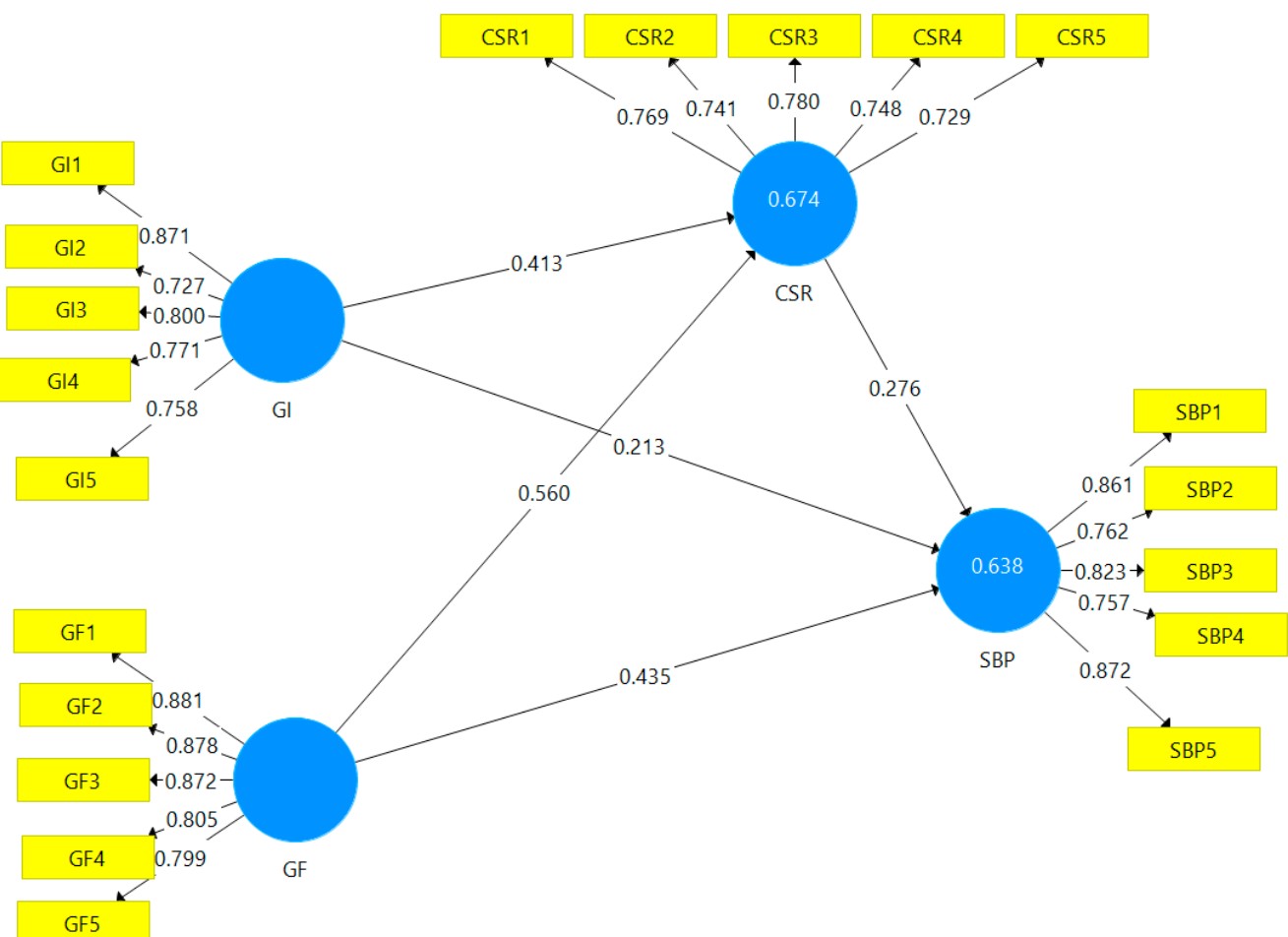

**Figure 2.** Measurement Model Assessment (output).

**Table 2.** Measurement model analysis.

| Constructs | Items | Factor Loading | Cronbach's Alpha | Composite Reliability | Average Variance Extracted (AVE) |
|---|---|---|---|---|---|
| Green Investment | GI | 0.871 | 0.851 | 0.89 | 0.619 |
| | GI2 | 0.727 | | | |
| | GI3 | 0.800 | | | |
| | GI4 | 0.771 | | | |
| | GI5 | 0.758 | | | |
| Green Financing | GF1 | 0.881 | 0.902 | 0.927 | 0.719 |
| | GF2 | 0.878 | | | |
| | GF3 | 0.872 | | | |
| | GF4 | 0.805 | | | |
| | GF5 | 0.799 | | | |
| Sustainable Business Performance | SBP1 | 0.861 | 0.874 | 0.909 | 0.667 |
| | SBP2 | 0.762 | | | |
| | SBP3 | 0.823 | | | |
| | SBP4 | 0.757 | | | |
| | SPB5 | 0.872 | | | |
| Corporate Social Responsibility | CSR1 | 0.755 | 0.813 | 0.869 | 0.57 |
| | CSR2 | 0.742 | | | |
| | CSR3 | 0.786 | | | |
| | CSR4 | 0.75 | | | |
| | CSR5 | 0.742 | | | |

Note: CSR = Corporate Social Responsibility, GI = Green Investment, GF = Green Financing, SBP = Sustainable Business Performance.

4.2.2. Discriminant Validity

The authors of this study evaluated the convergence and reliability using statistics before examining the discriminant validity [105]. Three techniques were applied to demonstrate discriminant validity in the Smart-PLS [106]. These are the analysis of the (HTMT) ratio, cross-loadings, and the Fornell–Larcker standard. First, we looked at the discriminant validity using the correlation. The criterion specifies that discriminant validity is guaranteed if the square root of the AVE for a certain concept is more significant than its relationship with other conceptions [106]. The squared AVE value in the bold diagonal of Table 3 implies that discriminant validity is not in danger because it is higher than the inter-construct correlation. This demonstrates a construct's uniqueness and explains concepts that other constructs in the model do not express [107].

**Table 3.** Fornnel–Larcker Criterion.

| | CSR | GF | GI | SBP |
|---|---|---|---|---|
| CSR | **0.754** | | | |
| GF | 0.729 | **0.848** | | |
| GI | 0.643 | 0.41 | **0.787** | |
| SBP | 0.73 | 0.724 | 0.569 | **0.817** |

Note: CSR = Corporate Social Responsibility, GI = Green Investment, GF = Green Financing, SBP = Sustainable Business Performance.

In the second step, cross-loading (CL) was employed to confirm the discriminant validity. According to this approach, a particular item should exhibit a higher correlation with its intended construct than with other constructs in the study. According to Table 4, the item has higher loadings than the others in the current investigation, showing a weak link between the variables and comprehensive discriminant validity. The final evaluation of discriminant validity is performed using the (HTMT) correlation ratio. If the route model includes constructs that are conceptually relatively similar, a threshold level of

0.90 is acceptable; however, if the HTMT value is higher than 0.90, it indicates a lack of discriminant validity. In our example, the HTMT values were less than 0.90, demonstrating the variables' sufficient discriminant validity and low correlations. Table 5 below displays the results.

**Table 4.** Cross Loadings.

|  | CSR | GF | GI | SBP |
|---|---|---|---|---|
| CSR1 | 0.769 | 0.352 | 0.674 | 0.564 |
| CSR2 | 0.741 | 0.359 | 0.538 | 0.486 |
| CSR3 | 0.78 | 0.48 | 0.532 | 0.541 |
| CSR4 | 0.748 | 0.713 | 0.384 | 0.595 |
| CSR5 | 0.729 | 0.725 | 0.324 | 0.549 |
| GF1 | 0.629 | 0.881 | 0.357 | 0.612 |
| GF2 | 0.696 | 0.878 | 0.399 | 0.685 |
| GF3 | 0.661 | 0.872 | 0.342 | 0.626 |
| GF4 | 0.523 | 0.805 | 0.301 | 0.502 |
| GF5 | 0.561 | 0.799 | 0.331 | 0.553 |
| GI1 | 0.495 | 0.394 | 0.871 | 0.481 |
| GI2 | 0.33 | 0.181 | 0.727 | 0.277 |
| GI3 | 0.451 | 0.174 | 0.8 | 0.303 |
| GI4 | 0.406 | 0.314 | 0.771 | 0.387 |
| GI5 | 0.694 | 0.428 | 0.758 | 0.601 |
| SBP1 | 0.618 | 0.528 | 0.488 | 0.861 |
| SBP2 | 0.489 | 0.532 | 0.372 | 0.762 |
| SBP3 | 0.631 | 0.735 | 0.464 | 0.823 |
| SBP4 | 0.599 | 0.51 | 0.473 | 0.757 |
| SBP5 | 0.628 | 0.618 | 0.516 | 0.872 |

Note: CSR = Corporate Social Responsibility, GI = Green Investment, GF = Green Financing, SBP = Sustainable Business Performance.

**Table 5.** Heterotrait and Monotrait Ratio (HTMT).

|  | CSR | GF | GI | SBP |
|---|---|---|---|---|
| CSR |  |  |  |  |
| GF | 0.826 |  |  |  |
| GI | 0.732 | 0.429 |  |  |
| SBP | 0.858 | 0.804 | 0.609 |  |

Note: CSR = Corporate Social Responsibility, GI = Green Investment, GF = Green Financing, SBP = Sustainable Business Performance.

### 4.3. Structural Model Analysis

Collinearity concerns evaluation is the initial step in structural model analysis. Hence, the potential for a collinearity issue was examined before looking at the structural relationship between latent variables. The study may exclude or combine predictor variables into a single construct when there is a collinearity issue. The presence of multi-collinearity issues was investigated in this work using the VIF. The constructs were free of collinearity problems, where collinearity statistics led to VIF values below the threshold of 5 and ranging from 1.51 to 2.86. Thus the results of VIF values are stated in Table 6 below as follow.

### 4.3.1. Path Analysis

With all variables observed, a system of equations allowing for multiple dependent variables was estimated using path analysis. Unlike regression models, path models can include multiple dependent variables. Smart-PLS was used to analyze the model, treating indicators as single-item constructs with equal weights. Significance testing for the path model was conducted using bootstrapping within Smart-PLS, which provides all necessary modeling and computation capabilities. The results were output immediately, and the process model is presented in Figure 3 and Table 7.

**Table 6.** Multi-Collinearity test results.

| Items | Variance Inflation Factor (VIF) |
| --- | --- |
| CSR1 | 1.94 |
| CSR2 | 1.733 |
| CSR3 | 1.937 |
| CSR4 | 1.66 |
| CSR5 | 1.618 |
| GF1 | 2.86 |
| GF2 | 2.642 |
| GF3 | 2.685 |
| GF4 | 2.036 |
| GF5 | 1.952 |
| GI1 | 2.653 |
| GI2 | 1.991 |
| GI3 | 2.011 |
| GI4 | 2.191 |
| GI5 | 1.51 |
| SBP1 | 2.609 |
| SBP2 | 1.739 |
| SBP3 | 1.968 |
| SBP4 | 1.682 |
| SBP5 | 2.725 |

Note: CSR = Corporate Social Responsibility, GI = Green Investment, GF = Green Financing, SBP = Sustainable Business Performance.

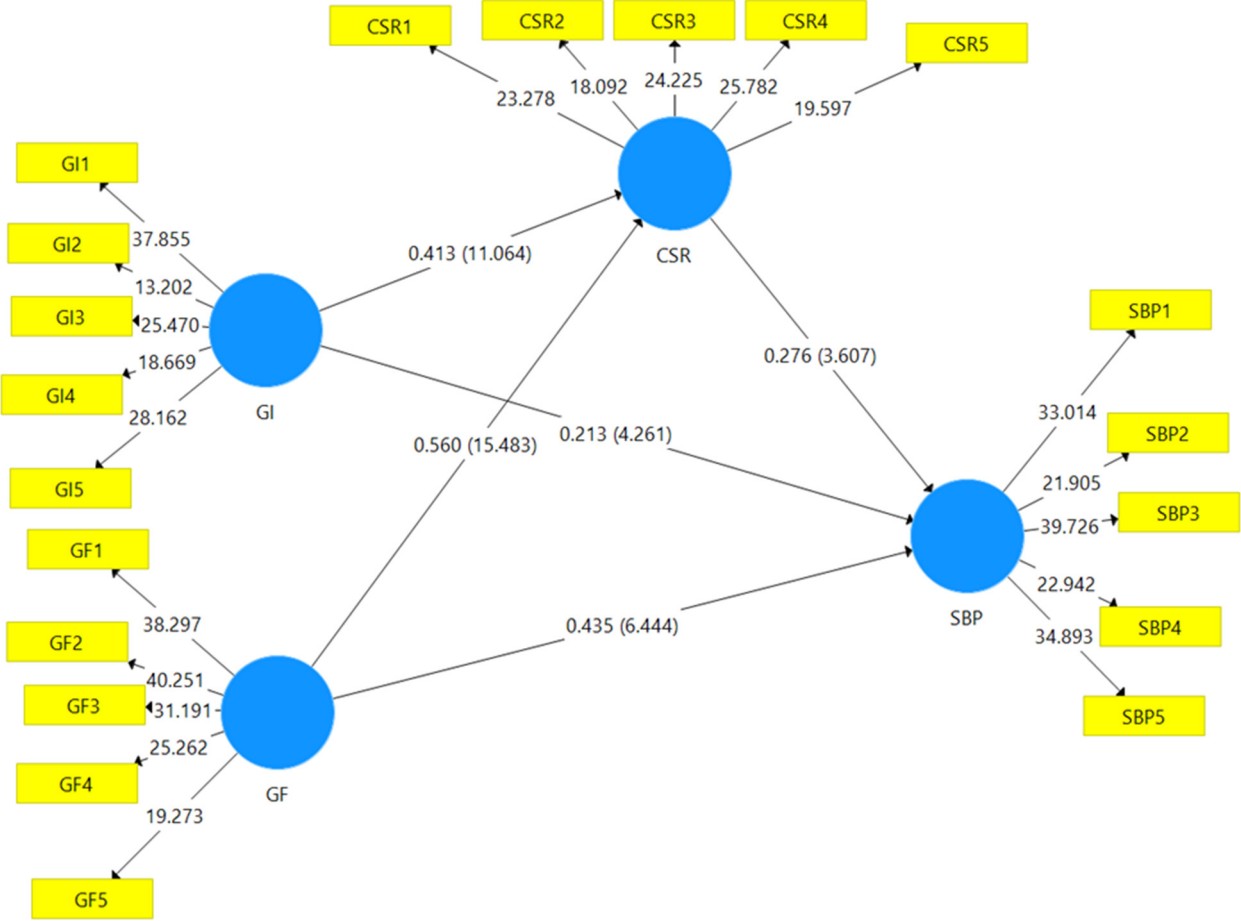

**Figure 3.** Structural model analysis.

**Table 7.** Path Analysis Results.

| Paths | Mean | STD | Coefficient | *t*-Value | *p*-Value |
| --- | --- | --- | --- | --- | --- |
| CSR → SBP | 0.279 | 0.076 | 0.276 | 3.604 | 0.000 |
| GF → CSR | 0.561 | 0.037 | 0.56 | 15.117 | 0.000 |
| GF → SBP | 0.433 | 0.068 | 0.435 | 6.414 | 0.000 |
| GI → CSR | 0.415 | 0.039 | 0.413 | 10.685 | 0.000 |
| GI → SBP | 0.213 | 0.051 | 0.213 | 4.196 | 0.000 |

Note: CSR = Corporate Social Responsibility, GI = Green Investment, GF = Green Financing, SBP = Sustainable Business Performance.

4.3.2. Hypothesis Testing

Once the validity and reliability of the outer models were confirmed and issues of multi-collinearity were addressed, the proposed relationships among the inner models were assessed. Table 6 displays the results of how the independent variables directly affect the dependent variable. The direction of the coefficient (positive or negative), its magnitude, and the *t*-statistic values are used to determine the level of significance of each path coefficient. Typically, *t*-values of 1.96 or greater fall within the ideal range of statistical significance. In this study, the significance of each structural path was determined using the bootstrapping method.

The first hypothesis (H1) postulates that green investment will enhance sustainable business performance (SBP). The study discovered a favorable and significant association between green investment and sustainable business performance, according to the scant evidence supplied thus far. It's critical to remember that without additional context and information, it is impossible to evaluate all of the finer points and findings of the study. Specifically, the study found that the beta coefficient for the relationship between GI and SBP was ($\beta = 0.213$, $t = 4.196$, $p < 0.000$), which suggests that the relationship is statistically significant.

Green investment has a positive and significant effect on the performance of sustainable businesses, as illustrated in Figure 3 and Table 8, respectively ($\beta = 0.213$, $t = 4.196$, $p < 0.000$). Therefore, the research supports H1, which posits that green investment contributes positively to sustainable business performance. It is worth noting that this finding should be interpreted in the context of the specific study and the methodology used and may not necessarily apply to all situations. Hypothesis (H2) claimed that green investment positively contributes to CSR. Based on the information you provided, the study found that green investment has a positive and significant effect on CSR. Specifically, the study found that ($\beta = 0.413$, $t = 10.685$, $p < 0.000$), for the relationship between green investment and CSR was 0.413, the *t*-value was 10.685, and the *p*-value was less than 0.000, which suggests that the relationship is statistically significant. Therefore, the research confirms H2, posing that green investment contributes positively to CSR. It is worth noting that this finding should be interpreted in the context of the specific study and the methodology used and may not necessarily apply to all situations. Additionally, it is important to note that the interpretation of the estimation value of a 41.3% increase in CSR for a 1-unit increase in green investment should be done cautiously, as it is based on a statistical model.

The second hypothesis (H3) proposes that green financing (GF) positively affects sustainable business performance. As demonstrated in Figure 3 and Table 8, the study findings indicate that green financing has a statistically significant and positive influence on sustainable business performance ($\beta = 0.435$, $t = 6.414$, $p < 0.000$). Thus, the research provides support for H3. In hypothesis H4, we claimed that green financing positively contributes to CSR. Thus, the results found a significant relationship between green financing and CSR, as demonstrated by the estimated values ($\beta = 0.56$, $t = 15.117$, $p < 0.000$). This implies that an increase in green finance results in an increase in CSR of 56% when all other variables are held constant. The research thus supports the accuracy of H4. The study's H5 hypothesis also postulated a correlation between CSR and improved sustainable company performance. The results show that CSR significantly and favorably influences sustainable

business success (β = 0.276, *t* = 3.604, *p* < 0.000). Holding other factors constant, a unit increase in CSR leads to a 27% increase in sustainable business performance. Therefore, the study supported the validity of H5. In summary, the results confirm all the direct hypotheses.

**Table 8.** Summarized Results Testing the Hypotheses.

| Hypotheses | Paths | β | *t*-v | *p*-v | BI (2.5%, 97.5%) | Decision |
|---|---|---|---|---|---|---|
| H1 (+) | GI → SBP | 0.213 | 4.196 | 0.000 | 0.114–0.31 | Supported |
| H2 (+) | GI → CSR | 0.413 | 10.685 | 0 | 0.335–0.488 | Supported |
| H3 (+) | GF → SBP | 0.435 | 6.414 | 0 | 0.297–0.564 | Supported |
| H4 (+) | GF → CSR | 0.56 | 15.117 | 0 | 0.48–0.626 | Supported |
| H5 (+) | CSR → SBP | 0.276 | 3.604 | 0 | 0.127–0.43 | Supported |
| | | | $R^2$ | | | |
| | CSR | SBP | | | | |
| | 0.67 | 0.64 | | | | |

Note: CSR = Corporate Social Responsibility, GI = Green Investment, GF = Green Financing, SBP = Sustainable Business Performance.

4.3.3. Mediation Analysis

The study conducted a mediation analysis to examine the potential mediating role of CSR in the relationships between green investment, green financing, and sustainable business performance. Table 7 demonstrates that the overall effect of green investment on sustainable business performance was statistically significant (H6: β = 0.327, *p* < 0.000). The relationship between green investment and sustainable business performance (SBP) persisted after controlling for a mediating variable (CSR) (β = 0.213, *p* < 0.05). In addition, statistically significant (β = 0.114, *p* < 0.000) was the indirect impact of green investments on sustainable company performance through CSR. The findings suggest that both the direct and indirect paths were statistically significant, suggesting that CSR had a role in mediating the relationship between green investment and sustainable company performance.

Similarly, the overall impact of green financing on the success of sustainable businesses was also statistically significant (H7: β = 0.59, *p* < 0.000), and even when CSR was present as a mediating variable, the relationship between green financing and successful businesses persisted (β = 0.435, *p* < 0.000). Additionally, statistically significant (β = 0.154, *p* < 0.000) was the indirect impact of green finance on the performance of sustainable businesses through CSR. The association between green financing and sustainable company performance was determined to be somewhat mediated by CSR since both direct and indirect channels were significant [108]. As a result, the study shows that CSR substantially mediates the connections between green investment, green financing, and sustainable company success and the results stated in Table 9.

**Table 9.** Mediation Analysis.

| Path | Total Effect | | Direct Effect | | Indirect Effect CSR | | | BI |
|---|---|---|---|---|---|---|---|---|
| | β | *p*-v | β | *p*-v | | β | *p*-v | (2.5%; 97.5%) |
| GI → SBP | 0.327 | 0.000 | 0.213 | 0.000 | GI → CSR → SBP | 0.114 | 0.001 | BI [0.11; 0.31] |
| GF → SBP | 0.59 | 0.000 | 0.435 | 0.000 | GF → CSR → SBP | 0.154 | 0.001 | BI [0.20; 0.56] |

Note: BI = bias corrected confidence interval, CSR = Corporate Social Responsibility, GI = Green Investment, GF = Green Financing, SBP = Sustainable Business Performance.

**5. Discussion and Implications for Policy**

Based on the stakeholder theory, the study's findings demonstrated a positive correlation between green investment, green financing, and sustainable business performance, implying that foreign investors and chemical company managers in Indonesia who prioritize allocating and utilizing substantial green investment and financial resources may

witness a sustained enhancement in the performance of chemical companies. Considering the interests of all stakeholders, including the environment, can enable businesses to achieve sustainable performance and generate value for society as a whole [13]. The research findings revealed a favorable association between green investment and sustainable business performance among foreign chemical companies operating in Indonesia.

Investing in environmentally sustainable initiatives can improve financial and non-financial outcomes, including improved profitability and return on assets and enhanced environmental and social responsibility. By committing to sustainable initiatives, foreign chemical companies in Indonesia can improve their image, attract socially responsible investors, and contribute to the country's efforts to reduce its carbon footprint [109]. The study suggests that foreign chemical companies in Indonesia should consider investing in eco-friendly production processes, waste reduction, emissions reduction, and sustainable supply chains to improve their overall business performance. These findings are consistent with those of another study [110].

The study found a significant positive relationship between green financing and sustainable business performance in the chemical industry. This suggests that companies prioritizing sustainability and investing in environmentally friendly initiatives will likely experience stronger financial performance over time [111]. Chemical firms should consider incorporating sustainability into their business strategies, which could lead to a competitive advantage and improve their financial outcomes. This could entail investing in renewable energy sources, cutting emissions and waste, and implementing more environmentally friendly production practices. Investors may also prioritize companies that prioritize sustainability and invest in green initiatives, resulting in a rise in demand for green financing and a shift in investment patterns towards more sustainable industries [112]. Measuring and reporting on sustainability performance is important in the chemical sector, as it can enhance companies' understanding of the outcomes of their sustainable initiatives and effectively communicate their progress to stakeholders. This can help establish trust with investors, customers, and other stakeholders, leading to a more sustainable and resilient industry. These findings are consistent with another study [113].

The investigation into the mediating function of CSR in the connections between green financing, green investments, and sustainable company performance. The relationship between green financing, investment, and sustainable business performance in Indonesia's international chemical industries is significantly mediated by CSR. The study results show that CSR acts as a significant mediator in the connection between green investment, green financing, and sustainable business performance in foreign chemical companies operating in Indonesia. These findings have important implications for these companies.

First, they emphasize the essential role of CSR in promoting sustainable business performance. Companies prioritizing CSR activities, such as environmental protection, social responsibility, and ethical business practices, will likely achieve better sustainability outcomes. Therefore, foreign chemical companies in Indonesia should concentrate on implementing CSR initiatives, including green investment and financing, to improve their sustainable business performance. Second, the mediation effect of CSR on the relationship between green investment and financing and sustainable business performance indicates that CSR initiatives play a crucial role in transforming green investment and financing into sustainable business outcomes. This finding implies that foreign chemical companies in Indonesia need to invest in green technologies and financing and prioritize implementing CSR initiatives to maximize the impact of these investments.

Finally, the study highlights the significance of comprehending the local context and regulatory environment while implementing sustainable business practices. Indonesia has specific environmental and social challenges that require customized solutions. Therefore, foreign chemical companies in Indonesia must know the local context and regulatory environment to implement sustainable business practices aligned with the country's sustainable development goals.

*5.1. Theoretical and Practical Implications*

5.1.1. Theoretical Implications

From a theoretical perspective, this study contributes to the literature on green investment, green financing, CSR, and sustainable business performance in several ways. First, this study supports the conceptual model based on stakeholder theory, adding to the scant literature, especially concerning foreign corporations operating in Indonesia. The sustainable business performance of foreign chemical companies in Indonesia is linked to several theoretical implications related to green investment, green financing, CSR, and sustainability.

First, green investment, green financing, and sustainable business performance relationship emphasize the role of environmentally sustainable projects and financial services in promoting sustainability. Green investment refers to investing in environmentally conscious initiatives, while green financing refers to providing financial services that support sustainable activities. Companies prioritizing sustainability and having access to green financing will likely perform better in environmental sustainability.

Second, CSR and sustainable business performance relationship suggests that responsible business practices help achieve long-term sustainability goals. CSR refers to a company's efforts to operate ethically and transparently and address social and environmental issues. Firms prioritizing CSR are more likely to invest in sustainable projects and practices, enhancing environmental performance and reputation [114]. Third, green investment, green financing, CSR, and sustainable business performance relationship highlights the importance of a comprehensive approach to sustainability management. Companies prioritizing sustainability in all operations, including investment, financing, and CSR, are more likely to achieve long-term sustainable business performance [115].

Adopting this approach necessitates a change in mindset from pursuing short-term profit maximization to prioritizing long-term sustainability objectives. In general, comprehending the connection between green investment, green financing, CSR, and sustainable business performance in the context of foreign chemical companies operating in Indonesia, which face significant pollution challenges, carries substantial theoretical implications for advancing sustainability. Firms must prioritize sustainability across all areas of their operations to achieve sustainable business performance over the long term.

5.1.2. Practical Implications

The practical implications for managers of foreign chemical companies in Indonesia are to invest in sustainable and environmentally friendly initiatives, prioritize corporate social responsibility, and ensure profitability and sustainability in the long run. This can be achieved by adopting sustainable production processes, waste reduction programs, and compliance with Indonesian environmental regulations. Managers must align their business practices with Indonesian expectations of responsible and sustainable practices, engage with stakeholders, and contribute to the community. Policymakers can support these efforts by providing incentives for sustainable practices, regulatory support, and penalties for non-compliance with environmental regulations. A shift towards sustainable business practices can improve the reputation of foreign chemical companies, attract socially conscious investors, reduce their carbon footprint, and benefit both the foreign chemical companies and the environment in Indonesia.

## 6. Conclusions and Direction for Future Research

This study has made several significant contributions to the existing literature in various ways. By employing CSR and the stakeholder theory, the research has examined the direct and indirect influences of green investment and green financing on sustainable business performance in foreign industrial firms in Indonesia. We proposed this research approach in light of the prior literature, and SEM was used to validate the hypotheses. This study offers a distinctive viewpoint on the sustainable business performance of green financing and investment. It examines the sample period during the current global warming

crisis, which can be likened to an economic downturn. In examining the effect of green investment and green financing on the sustainable business performance of foreign chemical industries operating in Indonesia, it's important to consider not only the environmental impact of these initiatives but also their social impact. A just transition to sustainability should prioritize the well-being of workers, communities, and other stakeholders affected by these industries, as well as the planet. Corporate social responsibility (CSR) can play a key mediating role in ensuring that green investments and financing are aligned with these goals. By incorporating social policies that address issues such as fair labor practices, community engagement, and human rights, foreign chemical industries operating in Indonesia can enhance their sustainable business performance in a way that benefits both society and the environment.

The empirical results highlight the significance of both the direct and indirect effects of green financing and investment on sustainable business performance. CSR also has a significant impact on mediating these connections. The importance and value of each element of green investment, green financing, and sustainable business performance were also stressed by CSR of the study. Implementing sustainable business performance requires connecting organizational vision to CSR using green resources. According to the current study, green finance should be adopted to achieve sustainable business performance because it has a greater positive impact on CSR than green investment. Financial institutions can also create and promote more green financing choices to accommodate the demand from businesses looking to fund their sustainable efforts. Additionally, the findings of this study could serve as benchmarks for future CSR and sustainable business performance.

*Limitations and Future Directions*

Despite the significant factors discussed in the preceding sections, it is equally vital to admit the restrictions and limits of this research that may aid future studies as restrictions provide further study and research directions. It is essential to acknowledge that the scope of the study is limited to foreign chemical industries in Indonesia, and the findings cannot be generalized to other countries or industries. Additionally, the study only examines the mediating role of corporate social responsibility and neglects other potential mediators that may impact the relationship between green investment, green financing, and sustainable business performance. Furthermore, the research relies on self-reported data from managers of foreign chemical companies, which may be prone to bias or inaccuracies. Last, the study does not explore the potential trade-offs or unintended consequences of green investment and green financing on other aspects of business performance, such as profitability or competitiveness. Future studies may look at data from additional industries and geographical areas to improve the generalizability of this approach. Different cultural, ethical, social, and environmental elements may influence the results differently, and the results may also differ in other ways. Because of time restrictions, a cross-sectional survey was used in this study; longitudinal research, however, could be conducted in the future for more precise and comprehensive findings.

**Author Contributions:** Developed the concept, J.Y. and E.D.; prepared the methodology, software, validation, formal analysis, research, resource, and data curation and wrote the first draft; J.Y. reviewed, edited, and managed the writing, E.D. All authors have read and agreed to the published version of the manuscript.

**Funding:** This research received no external funding sources.

**Institutional Review Board Statement:** Not applicable.

**Informed Consent Statement:** Not applicable.

**Data Availability Statement:** The data supporting this study's findings are available from the corresponding authors upon reasonable request.

**Acknowledgments:** The School of Management at the Wuhan University of Technology provided valuable support, and the authors express their gratitude to all those who participated in the study and provided precise data.

**Conflicts of Interest:** The authors declare no conflict of interest.

## Appendix A

**Table A1.** Measurements Construct Items.

| Variables | | Items | Source |
|---|---|---|---|
| Green Investment | (1) | Our company considers environmental preservation in its green investment decisions. | [99] |
| | (2) | Our company consistently makes green investments. | |
| | (3) | Our company engages in green initiatives to fulfill its obligation to the environment and society. | |
| | (4) | Our company finds the financial performance of green investment appealing. | |
| | (5) | Our company is not reducing its green investment to cut costs. | |
| Green Financing | (1) | Our company has established policies to ensure that financing is directed towards environmentally sustainable projects. | [100] |
| | (2) | Our company has allocated a specific budget for green projects and initiatives. | |
| | (3) | Our company has invested in green bonds or other similar financial instruments. | |
| | (4) | Our company has received financing from banks or other financial institutions for green projects. | |
| | (5) | Our company has engaged in advocacy or lobbying efforts to promote green financing at the national or international level. | |
| CSR | (1) | Our business participates in initiatives and campaigns that advance societal safety. | [48] |
| | (2) | Our business wants to grow sustainably while considering future generations' requirements. | |
| | (3) | Our corporation supports non-governmental organizations that operate in troubled regions. | |
| | (4) | Our business goes above and beyond what the law requires to protect customer rights. | |
| | (5) | Our company fully and promptly complies with all legal regulations. | |
| Sustainable Business performance | (1) | Our organization's net profit margin has increased. | [40] |
| | (2) | Our organization's return on investment has increased. | |
| | (3) | The growth of our profitability has been exceptional. | |
| | (4) | Our profitability has surpassed that of our competitors. | |
| | (5) | Our overall financial performance has outperformed our competitors. | |

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
