# Peer review of "The Effect of Green Investment and Green Financing on Sustainable Business Performance of Foreign Chemical Industries Operating in Indonesia: The Mediating Role of Corporate Social Responsibility"

_sustainability, doi:10.3390/su151411218_

Round 1

Reviewer 1 Report

In general, the assessed scientific study can be evaluated positively both in terms of content and form. Its scope 26 indicates that it is an attempt by the authors to comprehensively process the issue of green investments and their impact on the overall functioning of business companies, businesses, that is, entrepreneurs as such. As part of this question, we are talking about the investments of entrepreneurs - legal entities, which, as artificially constructed social entities, cannot act on their own, but this action is carried out through managers - natural persons. And here I miss, at least in the "2. Theoretical Background and Hypotheses Development" section, the definition of the person of the manager and his rights and obligations in relation to the entity he represents. This question is very appropriately treated by authors such as

Peráček T. & Kaššaj M. (2023). A Critical Analysis of the Rights and Obligations of the Manager of a Limited Liability Company: Managerial Legislative Basis. Laws. 12 (3):56. pp. 1-21. https://doi.org/10.3390/laws12030056

as well as

Stoic Andrea. (2020). THE VOLUNTARY DISSOLUTION OF A LIMITED LIABILITY COMPANY - A WAY OF ABUSING THE LAW? Perspectives of Law and Public Administration, 9 (1), pp. 65-69

I appreciate the establishment of very well formulated 5 hypotheses in chapter 2, to which I would expect an answer not only in section 4.3.2 but also in the conclusion itself.

Author Response

Dear Reviewer,

We would like to express our sincere gratitude for taking the time to read and review our article. Your insightful comments and questions have been invaluable in helping us improve the quality of our work.

Firstly, we appreciate your positive feedback on the clarity of our writing and the soundness of our arguments. Your encouragement has motivated us to continue striving for excellence in our writing.

We also appreciate your constructive criticism and suggestions for improvement. Your comments have helped us identify areas where we can strengthen our argument and provide more compelling evidence to support our claims.

In response to your specific questions and comments, we would like to address the following response in the next pages.

Once again, we sincerely thank you for your time and effort in reviewing our work. Your feedback has been instrumental in helping us refine our ideas and make our article more impactful.

Best regards,

Authors

Reviewer 2 Report

- Interesting and innovative paper with clear, action-oriented conclusions. The topic and geography are important and the tested hypothesis is relevant.

- The methodology needs to be improved, in particular to better define the terms of green investment, green finance, CSR and sustainable performance. The definitions come too late in the paper and remain somewhat vague. The paper needs more financial references

- The paper needs to better distinguish relationship between two variables and causation.

- The authors need to explain why this specific sector was chosen, how relevant it is for Indonesia and for FDIs, and whether the conclusions could be associated with other sectors of the economy.

More detailed comments:

[10] it is incorrect to say that previous studies have produced conflicting results regarding the relationships between green investment … and sustainable business performance. there is mounting and numerous evidence that ESG practices have a neutral or positive impact of corporates operating performance and valuation https://www.stern.nyu.edu/sites/default/files/assets/documents/NYU-RAM_ESG-Paper_2021%20Rev_0.pdf 

[11] previous studies have focused on industrialized countries, which actually have much higher pollution levels (not lower). what are those chemical industries, and what is the pollution that they cause,  be more specific

[15] chemical sector receives how much of total FDIs in indonesia? how relevant is the sector for FDI or for the economy? in other words, why did you pick this industry? are your conclusions relevant for other industrial sectors in Indonesia?

[23] green finance does not support only "companies with low energy use". the biggest potential is actually in companies that today have a meaningful negative impact and embark on a transition towards a low carbon economy. see Deschryver, De Mariz  What Future for the Green Bond Market  (2020) https://scholar.google.com/citations?view_op=view_citation&hl=pt-BR&user=J5cg0XEAAAAJ&citation_for_view=J5cg0XEAAAAJ:0EnyYjriUFMC.

[25] you may want to cite a better definition for green finance. the terms you mention are vague and dont explain the types of instruments (debt, equity, real assets) or why they are considered green (taxonomy, international best practices etc). green bonds are just a subset. In general, in your introduction, you may want to set the overall context of how finance can be directed towards sustainability (https://scholar.google.com/citations?view_op=view_citation&hl=pt-BR&user=J5cg0XEAAAAJ&sortby=title&citation_for_view=J5cg0XEAAAAJ:Zph67rFs4hoC). if you want to consider instruments, green bonds are the most well known debt instrument, but you want to convey the idea that there are other types. For example, blue bonds are a new feature within what the market calls “labelled bonds”. 

row 254. you mention “operational performance of foreign textile and apparel manufacturing enterprises”. the link with the chemical industry is not clear. 

row 314. consider if this is relationship or causation. “green investment has a favorable influence on CSR”. 

row 315. you need to define better what you mean by “sustainable business performance”. is the financial performance that is sustainable? how do you measure that financial performance? 

row 483. date of the study. who administered the questionnaires (the authors? is this public data?). who commissioned this study?

row 667. consider if this is relationship or causation. is it that businesses that receive more green financing become more financially sustainable, or is it that businesses that are successful are more inclined to ask for and attract green finance?

- Conclusion. Appropriate to include some thoughts about social policies and social impact, not just pollution and the green angle, in the context of a just transition and sustainability at large.

- The paper would benefit from a minor revision

Author Response

(The authors gave the same response as above.)

Reviewer 3 Report

Manuscript ID: sustainability-2456963

TypeArticle

TitleThe Effect of Green Investment and Green Financing on Sustainable Business Performance of Foreign Chemical Industries Operating in Indonesia: The Mediating Role of Corporate Social Responsibility

The paper is current, and the topic is relevant. The manuscript also aligns with the journal's scope.

According to the authors, in the second analysis stage, the study utilized a structural model to test the hypothesis and identify the structural relationships among the variables. The paper's mediation analysis also employed PLS-SEM to determine the direct and indirect effects. 

However, the study has some limitations that are not clearly disclosed by the authors.

Although some definitions are discussed in Section 2, there is no explanation about how the respondents (primary data collection) understood those terms. The authors need to clarify how they dealt with the data for fundamental reasons. Overall, the authors state that the paper 'used Smart-PLS and a structural equation model (SEM) to examine the gathered data and determine the causal link between green investment, green financing, CSR, and sustainable company performance.' However, there are several misunderstandings in the research design and aims.

Firstly, it is difficult to accept that a causal relationship is established since the authors do not provide any empirical strategy to address endogeneity problems. As the data is self-reported, the results are biased. Please provide more details about the causal estimations. Additionally, please present some robustness tests to validate the main results.

Secondly, the measurement of variables is based on respondents' perceptions, and it is not possible to infer a causal and relational effect from the results. Please explain how the authors addressed this limitation. 

Thirdly, explain how the authors ensured that the respondents had a consistent understanding of 'green investments' and 'green financing.' How can the authors guarantee that all respondents possessed the same knowledge of these terms? 

Fourthly, please provide more insights into the dependent variable. What is meant by 'sustainable company performance' (in some instances, the text refers to it as 'sustainable business performance')?

Line 508: The statement 'In this study, we used to measure the variables' is incomplete. Please provide additional information that clarifies the understanding. 

The questionnaire was created using a 5-point Likert scale (ranging from 1 for 'strongly opposed' to 5 for 'strongly agreeable'). The study employed 4 constructs with 23 measurement items to account for all independent, dependent, and mediator variables.

Regarding the dependent variable, please provide clear information on how it was defined and measured, specifically 'sustainable business performance.'

Concerning the independent variables, the authors mentioned 'green finance and investment' as independent variables that were quantified using green investment and financing indicators. However, the authors also mentioned the use of a Likert scale.

The section on limitations (6.1) does not sufficiently explain the meaning of the results." 

In general, one understands written English well. However, there are some typing errors (typos). For example, "futur" instead of 'future'.

Author Response

(The authors gave the same response as above.)

Reviewer 4 Report

The paper addresses an important and topical issue aiming at empirical investigation of the relationships between green investment, green financing, sustainable business performance and corporate social responsibility in the context of an emerging economy of Indonesia. Notwithstanding the originality of the research design and collected data, the following aspects of the study require further elaboration:

1)     The title of the paper suggests that the investigation is focused on ‘chemical industries’ whereas in line 254 the Authors mention that the examined sample covered ‘textile and apparel manufacturing enterprises’ – given the substantial differences between those industries (especially with respect to most common understanding of the term ‘chemical industry’ as the one that produces and develops industrial, specialty and other chemicals) the exact composition of the investigated sample should be consistently referred to throughout the text and its exact composition should be explicitly presented in subsection 3.2);

2)     In lines 253-254 the Authors state that ‘the essay focuses on the effects of green investment and marketing on…’ – given the scope of the paper it is not clear why they refer to ‘marketing’ instead of ‘green finance’?

3)     In line 273 the Authors argue that previous studies on green investment and sustainable business performance have focused 'only' on developed countries. Are there really no prior studies referring to that issue in the context of emerging economies, especially China (see e.g. https://doi.org/10.3390/su142315642), or even Indonesia (see e.g. https://doi.org/10.1051/e3sconf/20183109001)?

4)     Given the above the Authors should extend the literature review onto studies attempting to investigate similar research problems in the emerging economies;

5)     The specific items of the research questionnaire used to collect data in the area of 'Green Financing' (see Table 1A in Appendix A, p. 22) actually seem to be hardly related to that issue, as the term 'green financing' usually refers to specific methods of financing environmentally-friendly corporate investments and activities. Given the above, the data behind the variable GF employed in the study do not seem to correspond to the issue the Authors were actually trying to capture, which raises some serious doubts about the validity of the results and the conclusions derived on their basis;

6)     Overall, the adopted research design relies heavily on the subjective perception of the investigated issues by the examined respondents – it would therefore be highly advisable to enhance the analysis by introducing some more objective proxies of the analysed variables, e.g. measures and ratios based on hard data from annual or sustainability reports to make the research results more convincing. In particular, it seems crucial to employ some commonly accepted measures of financial performance such as ROA or ROE as proxies for ‘sustainable business performance – SBS);

7)     Some of the crucial elements of the general conceptualization of the undertaken research problem (see e.g. Figure 1, p. 10) should be thoroughly reconsidered, as it seems very likely that the relationships between both GI and GF and CSR may be bidirectional, i.e. that the orientation towards a more socially responsible business behaviour promotes both green investment and the use of green financing – adopting such perspective would, unfortunately, require some deeper reconstruction of the analytical part of the paper, especially the very design of the estimated structural equation model;

8)     Apart from employing the research questionnaire as the source of primary data the Authors mention the use of ‘secondary data’ extracted from ‘company sustainability reports, annual government reports, and policy documents’ (see lines 479-480) – given the description of the research resign provided in the paper it is not clear, however, how exactly these ‘secondary data’ were used;

9)     In line 319 the Authors open a new sentence with the phrase 'In other words,' however the claim that 'the performance of green firms is negatively affected by green finance' seems to be unrelated to conclusions of the study by Walley and Whitehead that they referred to directly before;

10)  The formulation of the third hypothesis of the study (H3 – see line 338) raises some doubts - why the Authors refer to a 'green enterprise' if the study is supposedly focused on 'polluting industries'? The same problem regards the fifth hypothesis (H5 – see line 379) where the Authors refer to ‘sustainable businesses’;

11)  In line 595 the Authors refer to bolded values in the diagonal of the matrix shown in Table 3, however in the current version of the paper the respective values seems to be displayed in standard font weight;

12)  As regards the results of CL analysis presented in Table 4 (p. 15) the Authors should comment on the value of cross loading between CSR4 and GF which is higher than the one observed for the parent construct (CSR).

 Although the quality of English in the manuscript is generally fine, some further proof-reading is advisable, see e.g.:

o   To avoid confusion I would recommend using the term ‘polluting industries’ instead of ‘polluted industries’ throughout the text.

o   In line 21 the phrase "these variables" seems somewhat imprecise - I would therefore recommend clarifying which variables are mediated by CSR.

o   In line 83 instead of ‘this industry’ I would suggest using ‘such industries’ given the content of the directly preceding sentence;

o   In line 89 I would suggest changing ‘in what extent’ to ‘to what extent’;

o   In line 126 instead of 'this investigation' I would recommend using 'investigation of...' as in the introductory sentence that opens a new section it would be better to state the investigated problem more explicitly.

o   In line 195 instead of 'banks' I would suggest using 'bank loans' or 'banking products';

o   The sentence ‘As a result, the above cause has not gotten attention to study there.’ requires correction with respect to style (lines 276-277);

o   In many cases the Authors refer to other studies as ‘a study’ (see e.g. line 363) or ‘the study’ (see e.g. line 350) without providing the names of their authors which seems somewhat confusing;

o   In line 415 instead of 'CSR mediates significant on' it should rather be 'CSR significantly mediates';

o   In lines 542-543 the Authors argue that 'Most respondents' age ranges were within a better range for the age group.' - apart from stylistic issues the term 'better range' seems debatable in this instance.

o   The style of the sentence in lines 604-606 needs adjustment.

o   In line 655 it is not clear what is meant by '+1'.

Author Response

(The authors gave the same response as above.)

Round 2

Reviewer 1 Report

I am glad that the authors incorporated all comments and therefore I recommend the article for publication in this form.

Reviewer 3 Report

The current version includes the suggested improvements.

Overall it is fine.

Reviewer 4 Report

In the revised version of the manuscript the Authors have addressed the majority of concerns raised in the first-round review which has allowed to significantly improve the overall coherence and scientific soundness of the paper.

Notwithstanding the above I would still recommend to have the paper thoroughly proof read by a native speaker to correct the remaining grammatical (see e.g. 'disagrees' in line 373), stylistic (see e.g. lines 247-248) and punctuation (see e.g. redundant hyphen in line 105) issues.